# Intestinal Population in Host with Metabolic Syndrome during Administration of Chitosan and Its Derivatives

**DOI:** 10.3390/molecules25245857

**Published:** 2020-12-11

**Authors:** Chen Yan, Cuili Zhang, Xuejiao Cao, Bin Feng, Xinli Li

**Affiliations:** Department of Biotechnology, College of Basic Medical Sciences, Dalian Medical University, Dalian 116044, China; yanchen94r@gmail.com (C.Y.); Biot_zhangcl@sina.com (C.Z.); sleetcao@dlmedu.edu.cn (X.C.); binfeng@dmu.edu.cn (B.F.)

**Keywords:** chitosan, chitosan oligosaccharides, gut microbiota, metabolic syndrome

## Abstract

Chitosan and its derivatives can alleviate metabolic syndrome by different regulation mechanisms, phosphorylation of AMPK (AMP-activated kinase) and Akt (also known as protein kinase B), suppression of PPAR-γ (peroxisome proliferator-activated receptor-γ) and SREBP-1c (sterol regulatory element–binding proteins), and translocation of GLUT4 (glucose transporter-4), and also the downregulation of fatty-acid-transport proteins, fatty-acid-binding proteins, fatty acid synthetase (FAS), acetyl-CoA carboxylase (acetyl coenzyme A carboxylase), and HMG-CoA reductase (hydroxy methylglutaryl coenzyme A reductase). The improved microbial profiles in the gastrointestinal tract were positively correlated with the improved glucose and lipid profiles in hosts with metabolic syndrome. Hence, this review will summarize the current literature illustrating positive correlations between the alleviated conditions in metabolic syndrome hosts and the normalized gut microbiota in hosts with metabolic syndrome after treatment with chitosan and its derivatives, implying that the possibility of chitosan and its derivatives to serve as therapeutic application will be consolidated. Chitosan has been shown to modulate cardiometabolic symptoms (e.g., lipid and glycemic levels, blood pressure) as well as gut microbiota. However, the literature that summarizes the relationship between such metabolic modulation of chitosan and prebiotic-like effects is limited. This review will discuss the connection among their structures, biological properties, and prebiotic effects for the treatment of metabolic syndrome. Our hope is that future researchers will consider the prebiotic effects as significant contributors to the mitigation of metabolic syndrome.

## 1. Introduction

Chitin is the second most abundant natural polysaccharide, serving as an important constituent of various creatures such as crustaceans [1], fungi [2], and insects [3], etc. Chitin, a biocompatible and biodegradable polymer, is widely applied in medicine [4], food preservation [5], etc. Chitin is a linear β-(1, 4)-linked polymer made up of N-acetylglucosamine (GlcNAc) units [6]. In nature, there are three different crystalline forms (α, β, and γ) of chitin. The hydrogen-bonding interactions of α-chitin in anti-parallel fashion can stabilize the aggregate structure and are hence widely exploited in biomaterials (e.g., food package films [7]). The compactness of β-chitin in parallel fashion is lower than that of α-chitin, facilitating the formation of biomaterials by crosslinking with other materials [8]. γ-chitin is a mixture of parallel and antiparallel sheets [9].

Even though chitin possesses the above-mentioned favorable functional feature, the poor solubility [10] limits its application in the pharmaceutical field. Researchers have turned their attention to chitosan (CS), a deacetylated derivative of chitin [11] (Figure 1). Chitosan is a random copolymer comprising d-glucosamine (the deacetylated ones) and *N*-acetyl-d-glucosamine units. The proportion of the d-glucosamine units defines the degree of deacetylation. A deacetylation degree reaching 50% implies the generation of chitosan; otherwise, the copolymer is still chitin [12].

Chitosan oligosaccharides (COS) are a mixture of oligomers of chitosan with an average molecular weight (MW) < 5000 Da; their structural units are also GlcNAc and 2-amino-2-deoxy-β-d-glucopyranose (GlcN) [13,14]. Chitosan oligosaccharide is readily soluble due to its short chain length, unlike chitin [15]. Chitosan and chitosan oligosaccharides have been widely investigated in recent reviews that have shown several crucial bioeffects, e.g., antibacterial [16,17], anti-inflammatory [16,18], and anti-diabetic [19,20] activities. These biological reactions are brought about by the physicochemical properties such as favorable water solubility and cationic nature [14].

Chitosan and its derivatives serve as prebiotics, nondigestible carbohydrates which selectively stimulate the growth of beneficial microbes [21]. Polysaccharides cannot be digested by digestive enzymes in the gastrointestinal tract but can be fermented by gut microbiota. The gut microbiome encodes the genome for hydrolyzing nondigestible carbohydrate. Additionally, different microbes containing different enzymes are capable of fermenting corresponding carbohydrates, which is a case of a symbiotic relationship between the host and the intestinal flora [22]. Furthermore, the degree of polymerization influences the location of fermentation in the gastrointestinal tract. Namely, soluble oligosaccharides tend to be fermented in proximal segments of the gastrointestinal tract, while less soluble polysaccharides tend to be fermented in the distal colon [22]. In general, polysaccharides and their derivatives have been advantageous to the gut microbiota.

Recent studies are focused on the effects of prebiotics on the composition of the gut microbiota. Prebiotics selectively promote the growth of beneficial enteric bacteria (e.g., *Lactobacillus*, *Bifidobacterium*, and *Faecalibacterium* [23]) and inhibit the occupation of harmful bacteria such as the genus *Desulfovibrio* [24] (phylum *Proteobacteria*). Polysaccharides with increasing populations of beneficial intestinal flora have been described in several recent reviews [22]. Concretely, oligosaccharides such as oligofructose and xylo-oligosaccharides could support the growth of beneficial bacteria, e.g., *Lactobacillus*, *Bifidobactetrium*, *Bacteroides* [25]. Furthermore, previous studies showed that chitosan and its derivatives could increase the population of probiotic genera *Bifidobacterium* [24], *Lactobacilli* [26], genus *Akkermansia* [27], and *Parabacteroides* [24].

There is a consensus that gut dysbiosis damages the intestinal barrier. For instance, mucin glycoprotein, especially mucin 2 (MUC2), produced by goblet cells to the lumen of the intestine, is dominant in the composition of intestinal mucus. It also constitutes the inner layer of intestinal mucus that contributes to the intact intestinal barrier and intestinal homeostasis [28]. A previous study has shown that mucins in the mucosa of germ-free mice can be attenuated due to gut dysbiosis, thus leading to the inflammation of the intestinal mucosa, increased mucosal permeability, and severe immune responses in lymphoid tissues [28]. Mucins compete with microbes (pathogens and commensals) for binding sites to the surface of the underlying epithelial lining, which prevents bacterial translocation across the intestinal barrier [29]. Mucins are exploited as a carbon source by the gut microbiota predominantly in the distal colon, hence promoting the growth of bacteria in the outer loose mucus and thickening the outer layer of intestinal mucus [29,30]. Therefore, mucin-degrading bacteria can decompose mucins into sugars and produce short-chain fatty acids (SCFAs), which offer energy to intestinal epithelial cells [31], thus contributing to the intact epithelial lining [32]. Recently, a mucin-degrading bacterium, *Akkermansia muciniphila*, has attracted considerable attention from many researchers due to its ability to lower blood lipids and blood glucose [31,33]. Recent evidence has suggested that the abundance of *Akkermansia muciniphila* in mouse models is inversely associated with obesity [33], insulin resistance [34], hepatic steatosis [35], and atherosclerosis [36], among others. Apart from the reduction of *Akkermansia muciniphila*, the dysregulation of the intestinal flora is closely corelated with diet-associated abnormalities [37]. The clustering of these pathologic conditions is categorized as metabolic syndrome [38], also known as syndrome X, a term depicting features that increase the risks of cardiovascular disease [38]. The diagnosis of metabolic syndrome requires at least three of the following clinical findings: abdominal obesity, increased fasting glucose, hypertension, high-density lipoproteins (HDLs), and hypertriglyceridemia [39]. Existing therapies for metabolic syndrome are mainly treatments for hyperlipidemia, hyperglycemia, hypertension, as well as dietary management and regular exercise [40]. There are three phases in a pathophysiological overview for metabolic syndrome: risk factors (overnutrition, physical inactivity, smoking, age, ethnicity, etc.); inflammation-induced insulin resistance, oxidative stress, mitochondrial dysfunction, etc.; type 2 diabetes, hypertension, dyslipidemia, non-alcoholic fatty liver disease (NAFLD) [41], etc. The pathophysiology of metabolic syndrome is associated with chronic inflammation in adipose tissue, vascular endothelium, etc., hence leading to cardiovascular diseases (e.g., atherosclerosis) [40].

In previous studies, in hosts with diabetes [42], hyperlipidemia, hepatic steatosis [43], atherosclerosis [36], and other clinical findings that are categorized under metabolic syndrome, it was observed that beneficial bacteria declined, intestinal mucus was attenuated, and harmful proinflammatory microbes increased in the intestinal mucosa. The prevalence of metabolic syndrome has displayed a worldwide growing trend: around one third of US adults have metabolic syndrome and the percentage of overweight and obese people in China showed a growth from 14.6% to 21.8% within 10 years. It is estimated that around one quarter of the world’s population (more than one billion people) suffers from metabolic syndrome [44]. Thus, it is a question of great interest to develop new therapies for metabolic syndrome [38].

Traditionally, type 2 diabetic patients take anti-diabetic drugs, especially metformin. Recent research supported the notion that metformin acted on the regulation of gut microbiota patterns by promoting the abundance of mucin-degrading *Akkermansia muciniphila* as well as various SCFA-producing microbes [45]. Chitosan and its derivatives play the same role in the intestinal microbiota. Recent studies reported several adverse reactions of metformin, such as metabolic acidosis and hyperlactatemia, because of obstructed elimination from the body, which would exert an additive effect to increase circulating metformin body levels [46]. There have been no chitosan-induced adverse reactions reported to date. Hence, chitosan and its derivatives have an underlying capacity for the attenuation of elevated fasting glucose.

Likewise, statins play an important role in lipid-lowering through alterations of the gut microbiota [47]. Recent studies illustrated that atorvastatin and rosuvastatin, mostly prescribed statins, induced significant changes in the increased abundance of genera *Bacteroides*, *Butyricimonas*, and *Mucispirillum* which are associated with decreased cholesterols and triglycerides [47]. Similarly, an increasing trend of the population of *Akkermansia muciniphila* was also found in atorvastatin-treated hypertension patients [48]. However, atorvastatin, the most prescribed medication, could induce muscle-related adverse reactions [49], myopathy, and even rhabdomyolysis. Additionally, the co-administration of statins and fibrates causes over 10-fold increased incidence of rhabdomyolysis [50], whereas the administration of chitosan and its derivatives can lower lipids in plasma through modification of the gut microbiota without noticeable adverse reactions [51]. Vitamin E, an antioxidant for the treatment for non-alcoholic fatty liver disease (NAFLD), was reported to influence gut microbiota. At phylum level, the population of *Proteobacteria*, comprising various pathogens such as *Escherichia coli* and *Salmonella*, was positively associated with increased consumption of vitamin E. The decreased abundance of *Verrucomicrobia*, including mucin-degrading *Akkermansia Muciniphilia*, was observed in vitamin-E-treated groups in mouse models. Recently, a quaternized chitosan derivative was found to have potent antioxidant effects [52]. Therefore, chitosan and its derivatives can act as an alternative in the treatment for NAFLD. Overall, chitosan and its derivatives may be applied as a potential adjuvant for metabolic syndrome treatment due to their innocuous properties.

This review will summarize alterations of the gut microbiota that have been recognized as either potential causal factors or associating factors of metabolic syndrome, consider the modulating effects of chitosan and its derivatives on metabolism syndrome by regulating gut dysbiosis, and explore the potential application of chitosan and its derivatives.

## 2. Biological Effects of Chitosan and Its Derivatives

Chitosan and chitosan oligosaccharides can be utilized to alleviate metabolic syndrome through alterations of the gut microbiota, reducing inflammation in the intestinal mucosa, and decreasing the risks of cardiovascular disease [53,54].

The bioeffects of chitosan and its derivatives come from the positively charged NH_3_^+^ group. The positive charge on the molecules neutralizes the negative charge on the proteins, lipids, and other substances with negatively charged radicals, which occupy critical positions in significant metabolism and other vital reactions [55]. Phospholipids on the surface of the membrane are negatively charged; hence, chitosan is capable of interacting with cells in multiple intercellular bioreactions [55]. At the molecular level, phosphate groups in deoxyribonucleic acid (DNA) and ribonucleic acid (RNA) have a negative charge [56], and proteins in extracellular fluid (pH > 7) are negatively charged due to carboxyl groups [57]. Therefore, chitosan can react to negatively charged cells and macromolecules. Molecular weight (MW) influences the efficiency of regulating lipid metabolism in chitosan-treated groups with high-fat diet (HFD) [58]. Previous studies showed that high-MW chitosan had a higher efficiency than low-MW chitosan in the inhibition of lipid absorption and enhancement of fatty acid oxidation in the liver [58]. High deacetylation reduces the chitosan–triglyceride interaction due to reduced intermolecular hydrophobic forces between *N*-acetyl-d-glucosamine units (acetylated groups) and triglycerides [59]. Therefore, an increased degree of deacetylation implies decreased oil-binding abilities.

### 2.1. Lipid-Lowering Effects of Chitosan and Its Derivatives

Hyperlipidemia and hyperglycemia, characterized by high serum cholesterol, triglycerides, or low-density lipoprotein (LDL), is an important risk factor associated with cardiovascular diseases [60,61].

Traditional lipid-lowering medication such as statins might mitigate hyperglycemia and hyperlipidemia through alterations of the gut microbiota [47]. Recent studies have shown that statins increase the abundance of beneficial microbes which are associated with decreased serum cholesterols and triglycerides [47]. Several muscle-related adverse events occurred after the administration of statins [49], whereas the administration of chitosan and its derivatives can lower serum lipid levels through the modification of the gut microbiota without noticeable adverse reactions [51]. Recently, the literature on chitosan-induced lipid-lowering effects has been reviewed.

The administration of chitosan reduced total cholesterol (TC) by 8%, low-density lipoprotein cholesterol (LDL-C) by 2%, and triglyceride (TG) by 19%, and it increased high-density lipoprotein cholesterol (HDL-C) by 14% [62]. Furthermore, chitosan has a positive influence on low-density lipoprotein (LDL) subtypes by increasing low-density lipoprotein-2 particles and decreasing atherogenic low-density lipoprotein. The low-density lipoprotein-2 particle is categorized as the large low-density lipoprotein, and the large size of low-density lipoprotein makes it less likely to be taken up by the arterial tissue than its smaller counterparts, suggesting less trans-endothelial transportation—namely, less atherogenic properties [62]. Chitosan can also inhibit the differentiation and the development of adipose cells by the gathering of triglycerides and expression of adipogenic epitopes. Furthermore, because cholesterol is a synthetic raw material for bile acids, chitosan can eliminate excess cholesterol and bile acids from the body by transferring them to the liver for excretion [3].

AMP-activated protein kinase (AMPK) plays a critical role in cellular metabolism in target tissues such as the liver, adipose tissue, and skeletal muscles [63], and it inhibits the deposition of fat by suppressing peroxisome proliferator-activated receptor-γ (PPAR-γ) and sterol regulatory element-binding proteins (SREBPs) [64]. In HFD hosts, Figure 2 illustrates that chitosan can activate AMPK and inhibit SREBPs and PPAR-γ [58]. Phosphorylation of AMPK induces the downregulation of adipogenic transcription factors (e.g., SREBPs and PPAR-γ), which lessens lipid synthesis and promotes lipolysis. PPAR-γ has promoted adipogenesis and triglyceride storage in adipocytes, while SREBPs can increase lipogenesis-related genetic expression such as fatty-acid-transport proteins, fatty-acid-binding proteins, fatty acid synthetase FAS, acetyl-CoA carboxylase, and HMG-CoA reductase [65,66].

Furthermore, Figure 2 illustrates decreased activity of lipoprotein lipase (LPL), reduced microsomal triglyceride transfer protein (MTTP) expression, and increased angiopoietin-like 4 (Angptl4) expression [58] in chitosan-treated mice.

Angptl4, a glycosylated and secreted protein, is also known to be an endogenous inhibitor of LPL and thereby alters plasma TG values. Angptl4 can be divided into several distinct regions: an *N*-terminal domain which contains two coiled-coil domains, a linker region, and a C-terminal fibrinogen-like domain [67,68]. Angptl4 protein is proteolytically cleaved by proprotein convertases (PCs) into different forms due to different patterns of the C-terminal and *N*-terminal domains. Angptl4 is distributed separately in various organisms, suggesting that the different patterns of Angptl4 may be associated with distinct species’ biological functions [68]. Additionally, full-length Angptl4 (flAngptl4, 50 kDa) and the yielded fragments, an *N*-terminal (nAngptl4) and a C-terminal (cAngptl4) fragment, indicate different functions in metabolism [69]. It was reported that the N-terminal helical region of Angptl4 is necessary to inhibit LPL [70]. In agreement with this notion, recent evidence has suggested that nAngptl4 and flAngptl4 inhibit LPL activities, although there is an absence of evidence showing the biological effects of cAngptl4 on lipid metabolism [69].

LPL, an enzyme attached to the luminal surface of endothelial cells in capillaries, is capable of decomposing TG and releasing free fatty acids, chylomicrons, and the very low-density lipoprotein (VLDL) particles [71]. Thereafter, increased free fatty acids can serve as PPAR-γ agonists and Angptl4 activators [72,73]. MTTP, an endoplasmic reticulum (ER)-resident protein, can transport lipids (e.g., TG, cholesteryl esters, free cholesterol, phospholipids, ceramides, and sphingomyelin), hence facilitating the optimal formation of lipoproteins [74]. Previous studies showed that the biological functions of absorbing and assembling chylomicrons into enterocytes are impaired in MTTP-IKO mice [75].

As shown in Figure 2, chitosan upregulated the Angptl4 protein expression, inhibiting LPL activity. Hence, reduced free fatty acids which come from VLDL and chylomicrons can inhibit the phosphorylation of PPAR-γ. Thereafter, chitosan promoted fat liberation and decreased fat storage. Consequently, plasma lipids and liver fat accumulation decreased due to the administration of chitosan [58].

### 2.2. Anti-Diabetic Effects of Chitosan and Its Derivatives

Insulin resistance, recognized as metabolic syndrome, is a contributing factor to the etiology of type 2 diabetes mellitus (T2DM) and is associated with a wide range of other pathogenic conditions including hypertension, hyperlipidemia, and others [76].

Type 2 diabetes mellitus, previously termed “noninsulin-dependent diabetes”, is a metabolic disease characterized by hyperglycemia [77]. Previous studies showed that the etiology of type 2 diabetes mellitus is related to β-cell dysfunction, chronic low-degree inflammation, and mitochondrial oxidative stress Patients with type 2 diabetes mellitus may initially not need insulin treatment to survive, even throughout their entire lifetime [77].

Traditionally, type 2 diabetic patients take anti-diabetic drugs, especially metformin. Recent research supported the notion that metformin modifies gut microbiota patterns by increasing the population of mucin-degrading *Akkermansia muciniphila* as well as various SCFA-producing microbes [45]. Chitosan and its derivatives impose similar effects on the intestinal microbiota without adverse reactions, whereas metformin was found to induce metabolic acidosis and hyperlactatemia because the obstructed elimination can exert an accumulative effect on plasma metformin levels [46]. Hence, chitosan and its derivatives have an underlying capacity to decrease fasting glucose.

In a recent randomized, controlled, double-blind, crossover clinical trial, the administration of chitosan oligosaccharides could reduce postprandial blood glucose levels in the 2-h oral sucrose tolerance test (OSTT), suggesting the improvement of impaired glucose tolerance [78]. Similarly, chitosan has reduced gluconeogenesis-related signals (e.g., PEPCK (phosphoenolpyruvate carboxykinase), p38, and AMPK), increased muscle glucose uptake-related signals (e.g., Akt), and promoted glucose transporter-4 (GLUT4) translocation from cytoplasm to membranes in streptozotocin-induced diabetic rats. Akt, known as protein kinase B or PKB [79], plays a critical role in various cellular processes. Phosphorylated Akt promotes glucose uptake and inhibits gluconeogenesis in response to insulin secretion [79,80]. Glucose uptake is predominantly mediated by the translocation of GLUT4 from vesicular intracellular compartments to cell membranes in skeletal muscles [79]. Phosphorylated Akt or AMPK activates PEPCK, a rate-limiting enzyme in gluconeogenesis, and the conversion of noncarbohydrates to glucose [80,81].

Therefore, chitosan is capable of inhibiting gluconeogenesis by inhibiting PEPCK and triggering glucose uptake in skeletal muscles through the translocation of GLUT4 from the cytoplasm to the cell membrane in diabetic rat models, thus lowering blood glucose [82].

### 2.3. Anti-Inflammatory Effects of Chitosan and Its Derivatives

Exogenous substances and tissue damage can initiate inflammation, followed by the generation of proinflammatory cytokines, immune cell recruitment and activation, and free radical production [83].

Nonsteroid anti-inflammatory drugs (NSAIDs) which inhibit cyclooxygenase (COX) are widely used as analgesics and antiplatelet agents. It was reported that approximately 12.8% of US adults (29.4 million) used NSAIDs at least three times a week for at least 3 months [84]. Chronic inflammation plays a critical role in various biological reactions (e.g., insulin resistance) and tissues (e.g., adipose tissue), triggering metabolic syndrome. However, NSAIDs can damage the intestinal mucosa and even cause alimentary tract hemorrhage [85]. Consequently, significant alterations of the gut microbiota in NSAID-treated hosts were found in recent research. Multiple in vivo and in vitro studies demonstrated that chitosan oligosaccharides could inhibit inflammatory reactions in response to lipopolysaccharide (LPS) or other stimuli [86] via AMPK phosphorylation [84]. Additionally, COS can inhibit TNF-induced (tumor-necrosis-factor-induced) NF-κB (nuclear-factor-κB) signaling, COX-2 and iNOS (inducible nitric oxide synthase) [84]. COX-2, a prostaglandin (PG)–endoperoxide synthase 2 enzyme, contributes to the production of prostanoid-like prostaglandin E2 (PGE2), which triggers inflammation [87]. iNOS is an enzyme that is responsible for nitric oxide (NO) synthesis.

### 2.4. Application of Chitosan and Its Derivatives in Clinical Trials

Previous clinical trial results make chitosan and its derivatives more likely to be used as alternative applications in the treatment of metabolic syndrome. In a randomized, double-blinded, placebo-controlled, crossover clinical trial, chitosan could lower total cholesterol (TC) levels and low-density lipoprotein cholesterol (LDL-C) concentrations in plasma [88]. In a recent clinical trial, after the administration of chitosan oligosaccharide (GO2KA1, a code name of chitosan oligosaccharide), it showed a decreasing trend in body fat ratios and waist circumferences, serum glucose levels, HbA1c (Hemoglobin A1c, a subtype of glycosylated hemoglobin), and C peptides (a byproduct in the formation of insulin) [89]. In addition, chitosan oligosaccharide (GO2KA1) could improve the impaired glucose tolerance and fasting glucose levels in a crossover, randomized, controlled clinical trial [78]. A meta-analysis indicates that chitosan treatment for patients significantly decreases DBP (diastolic blood pressure) in short-term interventions [90].

## 3. Alterations of Gut Microbiota by Chitosan and Its Derivatives

### 3.1. Gut Microbiota of Metabolic Syndrome

The gut microbiota has been previously identified in the pathogenesis of a variety of diseases, e.g., inflammatory bowel disease (IBD), colorectal carcinoma, Parkinson’s disease, etc. Diet and lifestyle are the most common causes of gut dysbiosis and metabolic profiles. Although the causal relationship between metabolism syndrome and gut dysbiosis remains uncertain, the close relationship between them is known [91]. The dysregulated intestinal microbiota could disturb gut homeostasis, resulting in gut barrier dysfunction and low-grade chronic intestinal mucosal inflammation with a concomitant translocation of bacteria and subsequent influx of inflammatory bacterial fragments into blood circulation. Such effects lead to increased lipopolysaccharide levels, also termed metabolic endotoxemia. Such extensive domino effects can lead to host obesity and insulin resistance [27].

Recent reviews have recognized diet as a dominant contributing factor to intestinal flora balances [91], suggesting that favorable alterations to the gut microbial profile through healthy diets (e.g., diets containing nondigestible carbohydrates) may prevent or alleviate pathological conditions which arise from a gut microbial imbalance. Dietary patterns are closely related to distinct constellations of bacteria in the gastrointestinal tract, termed enterotypes as well [92]. Recent studies have shown that the healthiest diet-related gut microbiota was closely associated with the highest microbial diversity [27]. To be specific, mice fed fish oil, a sort of unsaturated oil that possesses a suppressing effect on white adipose tissue (WAT) inflammation and related metabolic disorders, demonstrated an increased population of *Lactobacillus* and *Akkermansia*
*muciniphila***,** while mice fed lard, a sort of saturated oil, demonstrated a decreased population of *Bilophila* [93]. *Bilophila* has been previously identified to aggravate inflammatory colitis through white adipose tissue WAT inflammation via the toll-like receptor (TLR) signaling pathway [94,95] in genetically susceptible hosts [96]. *Akkermansia muciniphila* is capable of degrading mucins (Figure 3) which are located in the sterile inner layer of intestinal mucus, which is encompassed by a thicker outer layer filled with commensal bacteria [30]. Mucins are substrates for intestinal bacteria’s metabolic activities, as a result of its composition of amino acids and oligosaccharides. Consequently, the release of oligosaccharides, such as fucose, galactose, N-acetylglucosamine, N-acetylgalactosamine, sialic acid, sulfate, etc., can be further exploited by commensals, resulting in greater thickness of intestinal mucus [29,97].

In the one of the first studies that linked impaired gut microbiota to obesity, germ-free mice colonized with an obesity-related microbiota which was isolated from genetically obese ob/ob mice had an greater amount of body fat than those colonized with leanness-related microbiota [98]. Additionally, the intestinal flora contains glycoside hydrolases which are capable of digesting nondigestible carbohydrates and are also not found in the human genome, and it shows the potent capability of degrading and fermenting such dietary fibers [99,100]. Furthermore, such substrates supply energy to commensals and underlie intestinal epithelial cells via producing metabolites such as short chain fatty acids (SCFAs) [101].

It has been commonly recognized that obesity is closely associated with alterations of abundance ratios of *Firmicutes* to *Bacteroidetes*, which dominate the gastrointestinal microbiota and occupy over 90% of acknowledged phylogenetic categories [102]. The low-grade inflammatory reaction of obesity can be further aggravated by the influx of lipopolysaccharides (LPS), a component of bacteria with an inflammatory effect, resulting in endotoxemia. Endotoxemia is induced by gut dysbiosis and subsequent translocation of bacteria. Previous research showed a “dose-response relationship” between diet containing fat and increased plasma LPS levels [97]. Compared with the insulin and lipid profiles in the mice fed high-fat diet (HFD), the administration of LPS via subcutaneous infusions into mice could cause the same outcome: increased insulin resistance and obesity [94].

Previous research showed different components of gut microbiota in patients with type 2 diabetes mellitus compared to the gut microbiota in healthy groups [103,104], including a rising trend of the proportion of *Firmicutes* to *Bacteroidetes* in obesity hosts [98,105] and a reduced abundance of *Akkermansia muciniphila* in overweight children [106]. Type 2 diabetes mellitus with low-grade inflammation and dysregulation of gut microbiota induced further inflammation by increasing the remarkable influx of inflammatory agents, LPS, resulting in endotoxemia [107] as well, which aggravates insulin resistance [108]. A gut microbiota that contains a dominant population of *Proteobacteria* implies the aggravation of prediabetes. The abundance of *Proteobacteria* can be examined through DNA contents in blood (>85%) [109]. In previous research, patients with type 2 diabetes mellitus displayed a modified intestinal flora profile, characterized by a remarkable declining abundance ratio of butyrate-producing bacteria, e.g., *R. intestinalis* and *F. prausnitzii* [103]. Additionally, the diabetic gut microbial profiles were dominated by opportunistic pathogens such as *Bacteroides caccae*, various *Clostridiales*, *Escherichia coli*, and the sulfate-reducing species *Desulfovibrio* [103], which implies gut dysbiosis. The dysfunction of the diabetic gut microbial profile was related to increased transport of glucose through cell membranes, oxidative stress reactions, branched chain amino acid transport, sulfate reduction, and decreased butyrate synthesis [103]. Recent studies revealed that intragastric administration of *Akkermansia muciniphila* to HFD-fed mice in the absence of metformin attenuated impaired glucose tolerance and adipose tissue inflammation [110]. In line with this, a decreased population of *Akkermansia muciniphila* was observed in obese and type 2 diabetic hosts. Furthermore, the growth of *Akkermansia muciniphila* could be promoted by the administration of prebiotics and it was followed by improved plasma lipid and glucose profile, characterized by reduced weight gain, metabolic endotoxemia, adipose tissue inflammation, and insulin resistance [111]. Furthermore, *Akkermansia muciniphila* administration upregulated the intestinal levels of endocannabinoids which are involved in the regulation of metabolic homeostasis and gut barrier function through the microbiota–gut–brain axis [112]. In addition, it was reported that declined populations of phylum *Verrucomicrobiaceae* and genera *Akkermansia muciniphila* in the gastrointestinal tract could be linked to the metabolic profiles of patients with prediabetes [27].

Collections of digestive enzymes in the human body are devoid of hydrolases for degrading the glycosidic bonds of polysaccharides. However, commensals in the intestine containing hydrolases for decomposing polysaccharides can degrade polysaccharides. Numerous bacteria possess different sorts of enzymes for degrading polysaccharides. The main products of microbial fermentation of carbohydrates are short-chain fatty acids (SCFAs) and gases [22]. Chitosan oligosaccharides (COS) and the degradation of chitosan also have prebiotic-like effects in vitro [113] and in vivo [114,115]. Recent research has shown that the absorption of low molecular (low-MW) chitosan oligosaccharides (MWs of 13 kD and 22 kD) is higher than that of chitosan in the intestine, while high-MW chitosan oligosaccharides (230 kD) are non-absorbable in in vitro models [86]. It is postulated that low-MW COS can be absorbed directly in the intestine, whereas high-MW COS are fermented by the gut microbiota and produce SCFAs and other metabolites to maintain an intact intestinal barrier.

### 3.2. Gut Microbiota, Metabolic Syndrome and Chitosan and Its Derivatives

There is a general consensus that chitosan and its derivatives possess an antibacterial effect on pathogens. Such antibacterial effects are brought about by the positive charge on the NH3+ group of the glucosamine monomer in chitosan and its derivatives, interacting with a negatively charged microbial cell membrane [116]. To understand the specific mechanisms, several hypotheses are postulated.

The first hypothesis states that the electrostatic interactions between chitosan and bacterial surface induce the contact-induced lipid peroxidation via ROS (reactive oxygen species), and the production of lipid peroxide results in an impaired membrane and permeabilization of the bacterial cell membrane and subsequent leakage of intracellular substances, hence leading to bacteria death [116]. The second hypothesis posits that chitosan combines with nucleic acids and inhibits transcription and translation, thus resulting in cell death [117]. The third hypothesis refers to the chelation of vital metals and nutrients which could damage the cell membrane in bacteria, especially Gram-negative bacteria [117]. The fourth hypothesis refers to the formation of polymers fixed to the surfaces of bacteria, hence inhibiting the infusion of oxygen and the growth of microbes [118].

Additionally, metabolites such as short-chain fatty acids which come from the fermentation of chitosan and chitosan oligosaccharides are capable of boosting the growth of commensals (e.g., *Bifidobacterium* spp., *Bifidobacterium breve*, *Faecalibacterium prausnitzii*, and *Lactobacillus spp*. *Fusobacterium prausnitzii*, *Methanobrevibacter smithii*, and *Roseburia*) and the exclusion of pathogens (e.g., *S. aureus*) [34].

Recent studies demonstrated that chitosan oligosaccharides promoted weight loss, reduced plasma glucose levels, normalized insulin contents, and decreased the levels of triglycerides (TG), total cholesterol (TC), and low-density lipoprotein cholesterol (LDL-C) in hyperglycemic mice (type 2 diabetes mellitus model). In in vitro research in a high steatosis model (HepG2 cells) induced by oleic acids, chitosan oligosaccharides were observed to lessen the degree of fatty degeneration [119]. Furthermore, in Figure 3, it is demonstrated that chitosan oligosaccharides decreased the expression of HMGCR (hydroxy methylglutaryl coenzyme A reductase) and increased SMYD3 (SET and MYND domain containing protein 3) in mRNA and protein levels [119], as well as increasing the expression of CYP7A1 and GLP-1. HMGCR, the abbreviation of HMG-CoA Reductase, is the rate-controlling enzyme of the mevalonate pathway, which synthesizes cholesterol and other isoprenoids. The bioactivity of HMGCR was suppressed by chitosan oligosaccharides, resulting in reduced cholesterol levels. SMYD3, termed SET and MYND domain-containing protein 3 as well, is methyltransferase overexpressed in humanized tumor cells [120]. It enables the methylation of HMGCR, suppressing HMGCR. Glucagon-like peptide-1 (GLP-1) is an incretin hormone secreted by intestinal L-cells, which stimulates glucose-dependent insulin [32] production, suppresses glucagon secretion, and promotes the growth of β cells in pancreatic islets. GLP-1, probably associated with elevated insulin sensitivity, was increased by the administration of chitosan oligosaccharides. CYP7A1, cholesterol 7α-hydroxylase, is the rate-limiting enzyme of the classic synthesis pathway of bile acids (BA). CYP7A1 is critical to cholesterol catabolism and bile acid homeostasis. Thus, upregulated CYP7A1 reduces cholesterol levels [121]. Therefore, the upregulation of CYP7A1 can decrease cholesterols in plasma via the synthesis of bile acids in chito-oligosaccharide-treated hosts.

In addition, chitosan oligosaccharides reversed the decline of beneficial bacteria (e.g., phylum *Verrucomicrobia*) and the promotion of harmful bacteria (e.g., *Proteobacteria*) [119]. Furthermore, chitosan oligosaccharide treatment could decrease the microbial population of *Firmicutes*, *Streptococcus*, *Bacteroides fragilis*, *Clostridium coccoides*, *C. leptum* subgroup, and *Eubacterium rectale* in the gastrointestinal tract, especially *Escherichia coli*, a marker of gut dysbiosis [26]. Since various additional carbohydrate-degrading enzymes were encoded in the *Bacteroidetes*, compared with *Firmicutes*, the administration of chitosan oligosaccharides supplied an excellent growing environment for *Bacteroidetes* rather than *Firmicutes* [26]. In addition, at genus level, administration of chitosan oligosaccharides suppressed the population of *Lachnospiraceae* NK4A136 group, *Alistipes*, *Helicobacter*, *Ruminococcus*_1 and *Odoribacter* and upregulated the growth of *Lachnospiraceae*_UCG_001, and *Akkermansia*. It has been observed that *Helicobacter pylori* infection could be linked to insulin resistance, and the *Lachnospiraceae* NK4A136 group could serve as a distinctive marker of gut dysbiosis [122,123]. Previous research demonstrated that *Akkermansia* was positively correlated with serum insulin, while *Alistipes*, *Odoribacter*, and *Ruminococcus* were positively correlated with diabetic clinical manifestations (e.g., elevated glucose, insulin resistance, weight gain) through increased inflammatory responses via increasing TNF-α, monocyte chemoattractant protein-1 (MCP-1), and CD11c. Inflammation results in impaired gut barrier by inhibiting occluding, which constitutes cell tight conjunction, and suppressing the expression of AMP-activated protein kinase (AMPK) [124]. Recent studies have shown that AMPK activator AICAR improves intestinal epithelial development and barrier function by promoting epithelial differentiation and tight junction formation in Caco-2 cells which came from a colon carcinoma [125], suggesting the positive effects of AMPK on intestinal barrier.

Recently, bioinformatic analysis based on 16S rRNA sequencing has been a useful tool to identify the effects of multiple factors on the gut microbiota. For example, Zheng et al. have utilized Phylogenetic Investigation of Communities by Reconstruction of Unobserved States (PICRUSt) analysis to demonstrate the association of metabolic biomarkers and relative abundance of bacterial genera among groups. It was observed that the increased mobility of bacteria displayed in diabetic mice could be decreased by the administration of chitosan oligosaccharides according to the functional genes, and the oxidative stress of the gut microbiota by examining peroxisome, nucleotide excision repair, and protein processing in the endoplasmic reticulum occurred as well [124]. In general, chitosan may reverse the abnormal metabolic profiles characterized by clustering of clinical findings in metabolic syndrome via normalizing the imbalanced gut microbial profiles.

## 4. Conclusions and Future Perspectives

Metabolic syndrome has become a problem in most countries due to its increase of the prevalence of cardiovascular disease. Chitosan and its derivatives have shown mostly similar modifications of the gut microbiota to traditional anti-diabetic, lipid-lowering, and anti-inflammatory medication. The chitosan-added supplementary therapy also contributes to the modulation of abnormal metabolic profiles in the absence of adverse reactions, unlike conventional medication. Chitosan and chitosan oligosaccharides derived from chitin, the second most abundant polysaccharides, are potential therapeutic agents due to their wide range of bioactivities. Chitosan and chitosan oligosaccharides possess the hypolipidemic, anti-diabetic effect, accompanied by promotion of the growth of commensals in the gastrointestinal tract, which implies a positive correlation between the normalized metabolic profile and the improved gut microbial profile during the administration of chitosan and its derivatives. The hypolipidemic properties of chitosan and its derivatives increase the count of large-sized, low-density lipoprotein particles, activate AMPK phosphorylation, downregulate adiposity-related gene expression (e.g., SREBP-1c, FAS, HMGR), and suppress the PPAR-γ signaling pathway involved in the differentiation of adipocytes. The hypoglycemic effects of chitosan and its derivatives ameliorate abnormal glucose and insulin profiles via the inhibition of PEPCK and the translocation of GLUT 4 from cytoplasm to cell membranes. The anti-inflammatory properties of chitosan and chitosan oligosaccharides decreased proinflammatory cytokine production.

Chitosan and its derivatives exert corresponding effects to improve the decreased population of beneficial bacteria (e.g., *Lactobacilli*, *Bifidobacterium*, *Akkermansia muciniphila*) and reduce the elevated population of opportunistic pathogens (e.g., *Escherichia coli*) and inflammation-related bacteria (e.g., *Alistipes*, *Helicobacter*), considering the normalized glucose, insulin, triglycerides, cholesterol, and LDL-C.

Admittedly, it is unclear whether the regulatory effect on the gut microbiota during the administration of chitosan and its derivatives has a causal influence on the improved metabolic profiles in hosts with metabolic syndrome. Chitosan and its derivatives can be considered as a potential, promising, and effective supplementary therapeutic application for metabolic syndrome. The precise mechanisms of the promotion of the population of beneficial bacteria and the declined glucose and lipid levels need to be further explored in subsequent research.

The double role of chitosan as a food additive and adjuvant therapy makes it an excellent candidate for the formulation of functional foods and medications. Chitosan is not yet approved for clinical use in the treatment of metabolic syndrome, but for food additives and biological materials in Food and Drug Administration (FDA). In the drug development process, phase III clinical trials—namely, further large-scale, well-designed randomized controlled trials—are urgently needed. In addition, some gaps in knowledge still exist regarding the toxicity of chitosan. The correlation of molecular weight and deacetylation with the toxicity of chitosan is indistinct due to the contrasting conclusions of several investigations. Studies of the mutagenicity and genotoxicity of chitosan are limited. Therefore, the toxicity of chitosan requires further investigation. Our hope is that this review of chitosan and its derivatives will provide a significant new perspective that can help to propel this field to clinical translation.

## Figures and Tables

**Figure 1 molecules-25-05857-f001:**
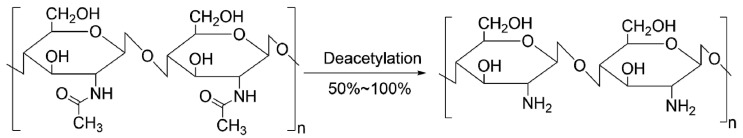
Structure of chitin and chitosan *N*-acetyl-d-glucosamine units converted to d-glucosamine through deacetylation. When the percentage of d-glucosamine to total monomers reaches or surpasses 50%, we can define the copolymer chitosan; otherwise, it remains chitin.

**Figure 2 molecules-25-05857-f002:**
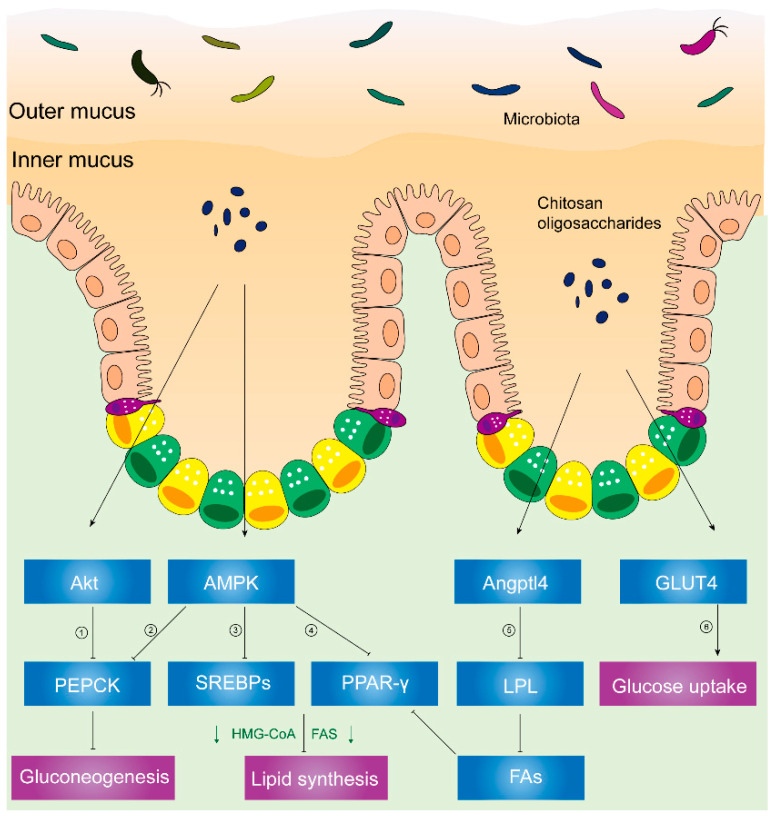
Chitosan and its derivatives mediate lipid and glucose metabolisms. ① Phosphorylated Akt can activate PEPCK, a rate-limiting enzyme in gluconeogenesis, and the conversion of noncarbohydrates to glucose. ② Activated AMPK can phosphorylate PEPCK. ③ Phosphorylated AMPK can inhibit expression of SREBPs, leading to inhibited HMG-CoA, FAS, etc. HMG-CoA is responsible for synthesis of cholesterol. FAS acts on synthesizing fatty acids. ④ Activated AMPK inhibits PPAR-γ, causing suppressed HMG-CoA, FAS, etc. ⑤ Upregulated Angptl4 inhibits activity of LPL, leading to less fatty acids released from VLDL and chylomicrons. Reduced fatty acids inhibit PPAR-γ, suppressing HMG-CoA, FAS, etc. ⑥ Translocation of GLUT4 from cytoplasm to skeletal muscle cell membranes can increase glucose uptake and decrease plasma glucose levels. **Abbreviations:** PEPCK, phosphoenolpyruvate carboxykinase; AMPK, AMP-activated kinase; SREBPs, sterol regulatory element-binding proteins; HMG-CoA, hydroxy methylglutaryl coenzyme A reductase; FAS, fatty acid synthetase; PPAR-γ, peroxisome proliferator-activated receptor-γ; Angptl4, angiopoietin-like 4; LPL, lipoprotein lipase; FAs, fatty acids; GLUT4, glucose transporter-4.

**Figure 3 molecules-25-05857-f003:**
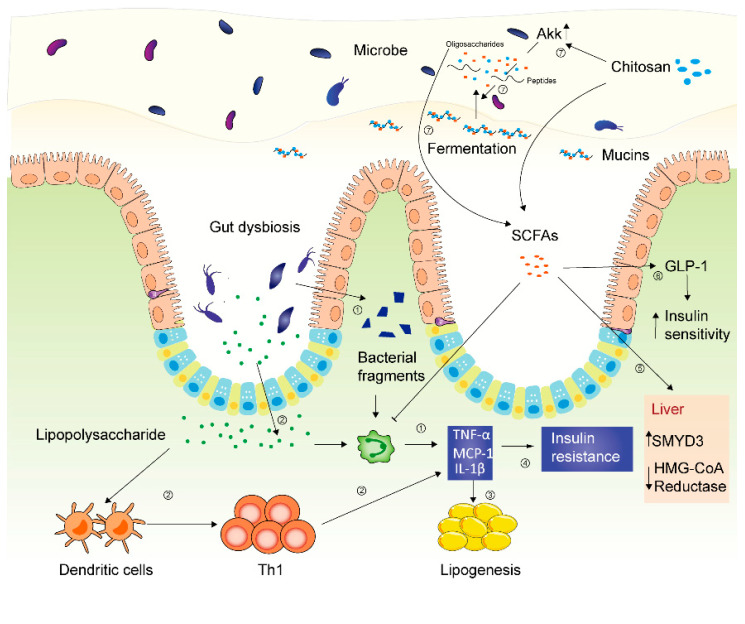
Mechanisms of chitosan in gut microbiota modulation in metabolic syndrome. ① Gut dysbiosis leads to bacterial fragment influx into blood circulation, inducing increased proinflammatory cytokines. ② Gut dysbiosis causes influx of lipopolysaccharides, which activates dendritic cells. Activated dendritic cells promote Th growth, resulting in increased proinflammatory factors (e.g., TNF-α, MCP-1, IL-1β). ③ Proinflammatory cytokines promote lipogenesis. ④ TNF-α induces insulin resistance through inhibition of sensitivity of insulin receptors. ⑤ SCFAs, metabolites of chitosan, inhibit HMG-CoA Reductase by overexpressing SMYD3. ⑥ SCFAs promote GLP-1, increasing insulin sensitivity. ⑦ Chitosan promotes the growth of Akk. Akk degrades mucins into peptides and oligosaccharides, which can be fermented by bacteria into SCFAs.

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
