# Peer review of "Intestinal Population in Host with Metabolic Syndrome during Administration of Chitosan and Its Derivatives"

_molecules, 2020, doi:10.3390/molecules25245857_

Round 1

Reviewer 1 Report

Manuscript ID: molecules-1023989

Type of manuscript: Review

Title: Intestinal population in host with metabolic syndrome during administration of chitosan and its derivatives.

Overview and general recommendation:

In the present work, the positive correlation between the alleviated conditions in metabolic syndrome hosts and the normalized gut microbiota in hosts with metabolic syndrome after the treatment of chitosan and its derivatives, implying that the possibility of chitosan and its derivatives to serve as the therapeutic application will be consolidated.

The paper deals with an exciting topic, which is the role of chitosan in metabolic syndrome. However, some issues need to be addressed before publication can proceed.

The abstract section is necessary to define all the abbreviations (Lines 13-15).

Check line 16. Absorbed to "be" accompanied?

It is necessary to highlight the novelty of the work. Why is it novel, and what is the real need that will be covered that has not been covered before?

Introduction

Lines 35-39. Why is this information relevant? Is it possible to connect the idea with some applications?

Figure 1: Some bonds look curved. Please check. The figure also highlights a 100% deacetylated chitosan. Is that the average? Otherwise, please modify the proper structure of the copolymer.

Lines 25-60 must be shortened. The related information to the main objective of the review begins in line 60.

Line 93: Check "mucosa[40]. ."

I suggest a section for the metabolism syndrome for educational purposes for readers with some illustrative schemes.

According to lines 143-145, the goals for the review are:

1) "summarize relevant literature including the modulating effect of the chitosan and its derivatives on metabolism syndrome through the mitigation of gut dysbiosis

2) highlight some specific alterations of intestinal flora that have been depicted as either potential causal factors or  associating factors for metabolic syndrome

3) explore the potential application of chitosan and chitosan oligosaccharides. However, I recommend that goal two be first described and then proceed with the chitosan influences over metabolic disorders.

Overall manuscript

Lines 148-155 are a repetition of the introduction section. It is not necessary to repeat the paragraph about chitosan properties.

Does only the Mw influence the regulation of lipid-metabolic capacity? What about the DD?

Lines 162-164 deserves more attention and explanation.

Any kind of chitosan is applicable for metabolism syndrome? What are the FDA requirements for clinical trials? Is it possible to separate a subsection only for clinical trials?

How can it be understood from Figure 2 the lipid and glucose metabolism? I believe a more graphical explanation is needed to understand lines 193-204, 224. Maybe some more description in the figure, using numbers for each event.

Line 266: A reference citation is needed.

Line 268: "are" is an unnecessary word.

Line 270 "a" not "an"

Line 274: In vivo and in vitro, not vivo and vitro.

I suggest adding a figure for the different mechanisms or roles of chitosan in Gut microbiota regulation.

It is highly required an-in depth English grammar review in all the manuscripts.

According to the literature reviewed, it should be added "future remarks or trends" for future research needed. Also, some legal requirements for pharmaceuticals interested in massive chitosan production could be interesting for clinical trials. Please add some information for legal requirements.

Author Response

Dear Doris You Assistant Editor (Molecules) Thank you very much for your kind suggestions on our manuscript. According to the comments of you and reviewers, we have revised our manuscript as follows: >>> Answer to the comments of reviewer #1: Question one: The abstract section is necessary to define all the abbreviations (Lines 13-15). Answer one: According to the reviewer’s suggestion. We have defined all the abbreviations in abstract in line 13 – 18. Question two: Absorbed to "be" accompanied? Answer two: We appreciate the reviewer’s kind suggestion. We have changed the sentence from “observed to accompanied” to “positively correlated to” in line 19. Question three: It is necessary to highlight the novelty of the work. Why is it novel, and what is the real need that will be covered that has not been covered before? Answer three: We appreciate the reviewer’s kind suggestion. We have added the novelty of the work in line 24-30. We have added that this review link the prebiotic-like effects of chitosan to its treatment for metabolic syndrome, which could be a novel insight to process the FDA approval. Question four: Lines 35-39. Why is this information relevant? Is it possible to connect the idea with some applications? Answer four: We appreciate the reviewer’s kind suggestion. We tend to highlight how chemical constitutions and structure bring about positive charge, while other polysaccharides are negatively charged. We have added some applications to α and β pattern of chitin in line 38-42. Question five: Figure 1: Some bonds look curved. Please check. The figure also highlights a 100% deacetylated chitosan. Is that the average? Otherwise, please modify the proper structure of the copolymer. Answer five: We appreciate the reviewer’s kind suggestion. We have rearranged the structure in figure 1 with Chemdraw. We have added the deacetylation degree “50%-100%” above the arrow in Fig. 1. The figure showed a deacetylated unit of chitosan. Chitosan is a random copolymer, and the percentage of the units in the copolymer define deacetylation degree. The deacetylation degree reaching 50% defines chitosan. The deacetylation degree fewer than 50% defines chitin. Question six: Lines 25-60 must be shortened. The related information to the main objective of the review begins in line 60. Answer six: We appreciate the reviewer’s kind suggestion. We have shorten the part from line 25 to 60. The shorten sentences are noted in line 34-48. Question seven: Line 93: Check "mucosa[40]. ." Answer seven: We appreciate the reviewer’s kind suggestion. We have changed from “the physical structure of mucosa” to “intact epithelial lining” in line 91. Question eight: I suggest a section for the metabolism syndrome for educational purposes for readers with some illustrative schemes. Answer eight: We appreciate the reviewer’s kind suggestion. We have inserted a section for metabolic syndrome in line 101-108. Question nine: According to lines 143-145, the goals for the review are: 1) "summarize relevant literature including the modulating effect of the chitosan and its derivatives on metabolism syndrome through the mitigation of gut dysbiosis 2) highlight some specific alterations of intestinal flora that have been depicted as either potential causal factors or associating factors for metabolic syndrome 3) explore the potential application of chitosan and chitosan oligosaccharides. However, I recommend that goal two be first described and then proceed with the chitosan influences over metabolic disorders. Answer nine: We appreciate the reviewer’s kind suggestion. We have described the goal two first and proceeded with the chitosan influences over metabolic disorders through gut microbiota, and mentioned the potential application of chitosan in line 145-148. Question ten : Lines 148-155 are a repetition of the introduction section. It is not necessary to repeat the paragraph about chitosan properties. Answer ten: We appreciate the reviewer’s kind suggestion. We have deleted the repetition part. The remained part begins in line 150. Question eleven: Does only the Mw influence the regulation of lipid-metabolic capacity? What about the DD? Answer eleven: We appreciate the reviewer’s kind suggestion. We have added that high DD (deacetylation degree) can reduce the oil-binding abilities of chitosan in line 163-166. Question twelve: Lines 162-164 deserves more attention and explanation. Answer twelve: We appreciate the reviewer’s kind suggestion. We have explained why chitosan can participate in sorts of biological reactions with the positive charge in line 156-160. Question thirteen: Any kind of chitosan is applicable for metabolism syndrome? What are the FDA requirements for clinical trials? Is it possible to separate a subsection only for clinical trials? Answer thirteen: We appreciate the reviewer’s kind suggestion. We have checked the FDA-Approved Drugs Database. Chitosan have not been approved of as drugs by FDA. Therefore, chitosan has not been in clinical use for the treatment for metabolic syndrome. We have added a subsection to show how chitosan works in clinical trials in line 286-297. As for FDA requirements, we have found that new drugs need to undergo the process of drugs development in < https://www.fda.gov/patients/learn-about-drug-and-device-approvals/drug-development-process>. There are four steps before the approval and one step after the approval: (1) Researchers discover new drugs, test potential benefits and pharmacokinetic-related information (e.g. absorption, distribution, excretion), figure out mechanisms and the optimal dose and form of drugs. (2) Preclinical researches: the toxicity of drugs need to be examined in vivo and in vitro before experiments on people. (3) Clinical research: drugs are examined on people in four phases of clinical trials (phase III finished before the approval). In phase I, 20-100 healthy volunteers or patients are included in experiments for safety. In phase II, several hundred patients are included in experiments for information about efficacy and side effects. In phase III, 300-3000 patients are included in experiments for information about efficacy and adverse reaction. The utilization of drugs in markets after approval is the phase IV. (4) FDA Drug Review: the FDA review team examine all submitted data and make a decision of approving it or not. (5) FDA Post-Market Drug Safety Monitoring: FDA would monitor applications of new drugs after approval. Question fourteen: How can it be understood from Figure 2 the lipid and glucose metabolism? I believe a more graphical explanation is needed to understand lines 193-204, 224. Maybe some more description in the figure, using numbers for each event. Answer fourteen: We appreciate the reviewer’s kind suggestion. We added several numbers in the figure 2 and added some description of information in line 193-204,224 below the figure. The information in line 193-204 (now in 189-198) are shown in 3, 4 in the figure 2. And the information in 224 (now in 214-232) are shown in 5 in the figure 2. Question fifteen: Line 266: A reference citation is needed. Answer fifteen: We appreciate the reviewer’s kind suggestion. We added a reference citation in line 272. Question sixteen: Line 268: "are" is an unnecessary word. Answer sixteen: We appreciate the reviewer’s kind suggestion. We have changed the sentence from “inflammation are triggered by exogeneous substances and tissue injury ” to “exogeneous substances and tissue damage can initiate inflammation” in line 270. Question seventeen: Line 270 "a" not "an" Answer seventeen: We appreciate the reviewer’s kind suggestion. We have changed “an critical role” to “a critical role” in line 276. Question eighteen: Line 274: In vivo and in vitro, not vivo and vitro. Answer eighteen: We appreciate the reviewer’s kind suggestion. We have changed from ”vivo and vitro ” to “in vivo and in vitro” in line 279. Question nineteen: I suggest adding a figure for the different mechanisms or roles of chitosan in Gut microbiota regulation. Answer nineteen: We appreciate the reviewer’s kind suggestion. We have added a figure 3 for the mechanism of chitosan in gut microbiota regulation in line 446. And figure 3 was mentioned in line 322, 408. Question twenty: It is highly required an-in depth English grammar review in all the manuscripts. Answer twenty: we appreciate the reviewer’s kind suggestion. We have revised the whole manuscript according to the revisions of TEXT (red mark). Question twenty one: According to the literature reviewed, it should be added "future remarks or trends" for future research needed. Also, some legal requirements for pharmaceuticals interested in massive chitosan production could be interesting for clinical trials. Please add some information for legal requirements. Answer twenty one: According to the literature reviewed, it should be added "future remarks or trends" for future research needed. Also, some legal requirements for pharmaceuticals interested in massive chitosan production could be interesting for clinical trials. Please add some information for legal requirements. We appreciate the reviewer’s kind suggestion. We have added “future perspective” for future research needed in line 496-506. We also search for documents in FDA and add information for FDA approval for chitosan in clinical use in line 497-500. With best regards. Yours sincerely, Xinli Li Chen Yan

Reviewer 2 Report

The review paper by Yan et al. summarized the biological interactions of chitosan with the biological systems. The current content is well organized and can give a clear picture of how chitosan influent host microbiota, which can be inspiring for the chitosan-based biomedicines.

The reviewer has no comment on the existing content, but suggest them include more discussions for the chitosan interaction with bacteria. Chitosan has been widely used to develop antibacterial materials, its interaction with the bacteria may also affect the host metabolism.

The authors should also comment on the degradation of chitosan as a nondigestible materials in human body.

Author Response

Dear Doris You

Assistant Editor (Molecules)

Thank you very much for your kind suggestions on our manuscript.

According to the comments of you and reviewers, we have revised our manuscript as follows:

>>> Answer to the comments of reviewer #2:

Question one: The reviewer has no comment on the existing content, but suggest them include more discussions for the chitosan interaction with bacteria. Chitosan has been widely used to develop antibacterial materials, its interaction with the bacteria may also affect the host metabolism.

Answer one: We appreciate the reviewer’s kind suggestion. We have added discussions on the chitosan interactions with bacteria in line 389-397.

Question two: The authors should also comment on the degradation of chitosan as a nondigestible materials in human body.

Answer two:   We appreciated the reviewer’s kind suggestion. We have added information of degradation of chitosan as prebiotics in human body in line 372-382.

With best regards.

Yours sincerely,

Xinli Li

Chen Yan

Round 2

Reviewer 1 Report

All my suggestions were properly addressed. An outstanding improvement in the manuscript could be evidenced. Under my criteria, the paper is ready for publication.